Quantitative assessment of fibroblast growth factor receptor 1 expression in neurons and glia

Choubey Lisha
Collette Jantzen C.
Smith Karen Müller karen.smith@louisiana.edu
Department of Biology, University of Louisiana at Lafayette , United States of America
Geloso Maria Concetta
Electronic publication date: 2017 Apr 18
Publication date: 2017
Volume: 5
Electronic Location ID: e3173
Received 2016 Aug 4; Accepted 2017 Mar 13
Copyright: ©2017 Choubey et al.
Copyright year: 2017
Copyright holder: Choubey et al.
License: This is an open access article distributed under the terms of the Creative Commons Attribution License, which permits unrestricted use, distribution, reproduction and adaptation in any medium and for any purpose provided that it is properly attributed. For attribution, the original author(s), title, publication source (PeerJ) and either DOI or URL of the article must be cited.
License URL: https://creativecommons.org/licenses/by/4.0/

Keywords: Astrocyte, SOX2, SVZ, Hippocampus, Cortex, Fibroblast growth factor, Doublecortin, Oligodendrocyte, FGFR1, Hypothalamus tanycyte

Funding: Brain Behavior Research Foundation Young Investigator Award (KMS) National Institute of Mental Health K01MH087845 This work was supported by the Brain Behavior Research Foundation Young Investigator Award (KMS), the National Institute of Mental Health (K01MH087845), Ray P. Authement College of Sciences start up funds from the University of Louisiana at Lafayette (KMS), and the Graduate Student Organization at the University of Louisiana at Lafayette (LC). There was no additional external funding received for this study. The funders had no role in study design, data collection and analysis, decision to publish, or preparation of the manuscript.

==============================
Background

Fibroblast growth factors (FGFs) and their receptors (FGFRs) have numerous functions in the developing and adult central nervous system (CNS). For example, the FGFR1 receptor is important for proliferation and fate specification of radial glial cells in the cortex and hippocampus, oligodendrocyte proliferation and regeneration, midline glia morphology and soma translocation, Bergmann glia morphology, and cerebellar morphogenesis. In addition, FGFR1 signaling in astrocytes is required for postnatal maturation of interneurons expressing parvalbumin (PV). FGFR1 is implicated in synapse formation in the hippocampus, and alterations in the expression of Fgfr1 and its ligand, Fgf2 accompany major depression. Understanding which cell types express Fgfr1 during development may elucidate its roles in normal development of the brain as well as illuminate possible causes of certain neuropsychiatric disorders.

Methods

Here, we used a BAC transgenic reporter line to trace Fgfr1 expression in the developing postnatal murine CNS. The specific transgenic line employed was created by the GENSAT project, tgFGFR1-EGFPGP338Gsat, and includes a gene encoding enhanced green fluorescent protein (EGFP) under the regulation of the Fgfr1 promoter, to trace Fgfr1 expression in the developing CNS. Unbiased stereological counts were performed for several cell types in the cortex and hippocampus.

Results

This model reveals that Fgfr1 is primarily expressed in glial cells, in both astrocytes and oligodendrocytes, along with some neurons. Dual labeling experiments indicate that the proportion of GFP+ (Fgfr1+) cells that are also GFAP+ increases from postnatal day 7 (P7) to 1 month, illuminating dynamic changes in Fgfr1 expression during postnatal development of the cortex. In postnatal neurogenic areas, GFP expression was also observed in SOX2, doublecortin (DCX), and brain lipid-binding protein (BLBP) expressing cells. Fgfr1 is also highly expressed in DCX positive cells of the dentate gyrus (DG), but not in the rostral migratory stream. Fgfr1 driven GFP was also observed in tanycytes and GFAP+ cells of the hypothalamus, as well as in Bergmann glia and astrocytes of the cerebellum.

Conclusions

The tgFGFR1-EGFPGP338Gsat mouse model expresses GFP that is congruent with known functions of FGFR1, including hippocampal development, glial cell development, and stem cell proliferation. Understanding which cell types express Fgfr1 may elucidate its role in neuropsychiatric disorders and brain development.

Introduction

Differential binding of Fibroblast growth factors (FGFs) to FGF receptor isoforms may confer a high degree of selectivity leading to signaling events that lead to a multitude of specific cellular responses (Chellaiah et al., 1999; Iwata & Hevner, 2009; Hebert, 2011; Ornitz & Itoh, 2015). Multiple Fgf ligands and three of the Fgfrs (Fgfr1, Fgfr2, and Fgfr3) are expressed in the CNS during development, in postnatal stem cell niches such as the dorsal VZ, and in glial lineages (El-Husseini, Paterson & Shiu, 1994; Belluardo et al., 1997; Bansal et al., 2003; Cahoy et al., 2008; Doyle et al., 2008; Garcia-Gonzalez et al., 2010; Azim, Raineteau & Butt, 2012). This complex system of Fgfs and Fgfrs plays a pivotal role in the normal development, maturation, and function of the central nervous system (CNS) (Iwata & Hevner, 2009; Stevens et al., 2010b; Hebert, 2011). FGF signaling is implicated in patterning of the CNS, in determining neuronal and glial cell fate, in influencing cerebral cortex size through maintenance of radial glial stem cells, in cerebellar development, and in regional patterning of the neocortex and midbrain-hindbrain boundaries (Vaccarino et al., 1999; Fukuchi-Shimogori & Grove, 2001; Korada et al., 2002; Storm, Rubenstein & Martin, 2003; Shin et al., 2004; Gutin et al., 2006; Smith et al., 2006; Storm et al., 2006; Cholfin & Rubenstein, 2007; Muller Smith et al., 2008; Kang et al., 2009; Stevens et al., 2010a; Rash et al., 2011; Müller Smith et al., 2012; Rash et al., 2013; Kang et al., 2014; Smith et al., 2014).

Loss of FGFR1 function by hGFAP-Cre-induced deletion of Fgfr1flox∕flox alleles in the dorsal telencephalon of mice results in decreased hippocampal size and volume, with a reduction in the number of dividing progenitor cells of the ventricular zone and DG (Ohkubo et al., 2004). Fgfr1 mutants also exhibit a disruption in corpus callosum and hippocampal commissure due to abnormal midline glia development (Smith et al., 2006; Tole et al., 2006). The midline glial cells fail to undergo soma translocation and formation of the indusium griseum leading to midline commissural axon guidance defects (Smith et al., 2006). Furthermore, these mice exhibit postnatal loss of maturation in GABAergic interneurons expressing parvalbumin (PV) and exhibit behavioral hyperactivity (Muller Smith et al., 2008; Smith et al., 2014). Hyperactivity and a decrease in number of interneurons in the cortex co-occur in patients with schizophrenia (Volk et al., 2000; Akbarian & Huang, 2006; Hashimoto et al., 2008; Volk & Lewis, 2013). Interestingly, Fgfr1 expression was found to be increased in the prefrontal cortex of individuals with schizophrenia (Volk, Edelson & Lewis, 2016). Dual inactivation of floxed alleles of Fgfr1 and Fgfr2 results in abnormal cerebellar morphogenesis including reduced size of the cerebellum due to a defect in proliferation of both cerebellar glia and granule cell precursors, abnormal orientation and morphology of Bergmann glia, and loss of laminar architecture (Müller Smith et al., 2012). This phenotype is similar to that observed in Fgf9 mutants (Lin et al., 2009). FGFRs are implicated in maintaining astrocytes in a non-reactive state, and in impeding glial scar formation (Kang et al., 2014). When Fgfr1 deletions were targeted to oligodendrocyte lineages, they did not disrupt oligodendrocyte birth, but modulated myelin sheath thickness and remyelination in chronic demyelination models (Furusho et al., 2012; Furusho et al., 2015). Administration of FGF2 into the lateral ventricles has also been shown to increase the number of oligodendrocyte precursor cells in the SVZ (Azim, Raineteau & Butt, 2012).

Patients with major depressive disorder (MDD) and bipolar disorder have altered gene expression of FGFs and FGFRs (Evans et al., 2004; Gaughran et al., 2006). In situ hybridization revealed that mRNA for Fgfr1, and its ligand Fgf2, were both down regulated in the hippocampus of rats that had undergone the social defeat paradigm (Turner et al., 2008). Microinjections of FGF2 into the lateral ventricles of rats resulted in an increase in Fgfr1 mRNA in the DG within 24 h post FGF2 injections and was accompanied by acute antidepressant-like effects in the force swim test (Elsayed et al., 2012). Furthermore, increased anxiety, dysregulation of the hypothalamic pituitary axis and decreased hippocampal glucocorticoid receptor expression is observed in FGF2 knockout mice. These effects are reversible by administration of FGF2 (Salmaso et al., 2016). FGF22 and FGF7 are presynaptic organizing molecules that promote differentiation of excitatory and inhibitory presynaptic terminals in the hippocampal cornu ammonis (CA) region 3 through combinatorial signaling of sets of FGFRs (Umemori et al., 2004; Terauchi et al., 2010; Dabrowski et al., 2015). Given the data that FGF2/FGFR1 signaling is important for the regulation of mood and affect, and that FGFR1 signaling may participate in synaptogenesis, a better understanding of the cell types expressing Fgfr1 is important to improving our understanding of its actions.

Previous estimates of Fgfr1 expression have been derived from in situ hybridization studies and from immunochemistry using antibodies against FGFR1 (Gonzalez et al., 1995; Belluardo et al., 1997; Bansal et al., 2003; Ohkubo et al., 2004; Blak et al., 2005), while more recently, molecular studies have identified the timing and or cell specific expression of Fgfr1 (Doyle et al., 2008; Garcia-Gonzalez et al., 2010; Azim, Raineteau & Butt, 2012). In embryonic mice, Fgfr1 is strongly expressed in the hippocampal hem, choroid plexus, cortical ventricular zone, and cortical midline (El-Husseini, Paterson & Shiu, 1994; Bansal et al., 2003; Ohkubo et al., 2004; Smith et al., 2006). In adult mice, the strongest Fgfr1 expression is observed in the hippocampus (Ohkubo et al., 2004). Based on a literature review, Turner, Watson & Akil (2012a) surmised that neuronal populations in the adult hippocampus and cortex mostly express Fgfr1, in contrast to other FGF receptors that are considered to be expressed primarily in glia.

Despite clear advances in our understanding of FGF signaling derived from in situ hybridization, it suffers from poor cell-type resolution. Likewise, although immunocytochemistry using antibodies raised to FGFR1 has proven to be important, cross-reactivity to other FGFRs remains a concern. To circumvent these issues, we investigated Fgfr1 expression in PV+ interneurons, employing a transgenic reporter line, tgFGFR1-EGFPGP338Gsat bacterial artificial chromosome (BAC), that was obtained from GENSAT, http://www.gensat.org (Smith et al., 2014). In this transgenic line, the gene encoding enhanced green fluorescent protein (EGFP) is regulated under the same promoter as Fgfr1. Thus, GFP fluorescence should indicate expression of genes encoding Fgfr1. We previously showed that in tgFGFR1-EGFPGP338Gsat mice, PV+ interneurons did not colocalize with GFP+ cells. Thus, the decrease in PV+ interneurons due to inactivation of Fgfr1 occurs non-cell-autonomously (Smith et al., 2014). We also observed that a large number of glia appeared to express Fgfr1 in adult mice. In the present study, we extend our studies and present a quantitative analysis of the relative expression of Fgfr1 in neurons versus glial cells during postnatal development of the telencephalon. We show that Fgfr1 is differentially expressed, primarily in GFAP+ astrocytes and OLIG2+ cells, with a minority of cells colocalizing with NeuN+ neurons. Furthermore, SOX2+ cells, BLBP+ cells and DCX+ cells in the cortex, hippocampus, subventricular zone (SVZ), and hypothalamus of mice are colocalized with the GFP signal, indicating that these cells also express Fgfr1.

Methods and Materials

Animals

Wild type Swiss Webster mice were crossed with mice expressing enhanced green fluorescent protein (EGFP) under the same promoter as Fgfr1 (tgFGR1-EGFPGP338Gsat). The transgenic line, tgFGR1-EGFPGP338Gsat was generated from the GENSAT project (GENSAT.org) by microinjecting bacterial artificial chromosome with Fgfr1 promoter driving EGFP into the pronucleus of fertilized mouse eggs. GENSAT produces transgenic BAC-EGFP reporter and BAC-Cre recombinase driver mouse lines with the aim to map the expression of genes in the CNS of mice (Heintz, 2004). This line was obtained from the Mutant Mouse Resource Center (MMRRC.org) at UC Davis. This study was conducted in an ethical manner, utilizing the recommendations of The Guide for the Care and Use of Laboratory Animals of the National Institutes of Health. Animals were euthanized under the University of Louisiana at Lafayette IACUC committee APS numbers 2012-8717-046, 2013- 8717-053, 2014-8717 040, 2015-8717-033. Tissue collected for this study was performed under ketamine/xylazine cocktail for P7 and older animals.

Genotyping

The mice were genotyped by polymerase chain reaction (PCR) for GFP and through GFP screening with goggles containing a GFP filter (BLS LTD). For PCR based genotyping, tails of mice were collected and DNA was extracted from the tail using 50 mM sodium hydroxide (95 °C for 30 min), followed by neutralization with 1M TRIS (pH 7.6). Master mix for 1 reaction of PCR for amplifying GFP was created using 2.5 µl of 10× PCR buffer, 0.5 µl of 10 mM dNTP mix, 1 µl of eGFP forward and reverse primer mix (Forward: AAGTTCATCTGCACCACCG and Reverse: TGCTCAGGTAGTGGTTGTCG), 0.2 µl of 5 units/µl Hot start Taq Polymerase and 18.8 µl of distilled water. Two microliters of DNA sample were added to 23 μl of Master mix per PCR tube and the samples were amplified in the Applied Biosystems 96 Well Thermocycler. In adult tgFGR1-EGFPGP338Gsat positive mice, we found we could reliably genotype mice due to GFP expression in the eye, using GFP goggles (available from BLS).

Immunostaining

P7 mice were anesthetized with ketamine/xylazine (100 mg and 10 mg per kg of body weight respectively) and euthanized by cervical dislocation. Brains were dissected out and in 4% PFA overnight. Adult mice were euthanized by cardiac perfusion under deep anesthesia (ketamine/xylazine as above). Mice were perfused with cold 1× Phosphate Buffered Saline (PBS) followed by 4% PFA in 1× PBS. Brains were post fixed overnight and cryoprotected as described above. Brains collected from P7 and one month ages were cryoprotected in 20% sucrose/1xPBS, and cryopreserved with exposure to dry ice and embedded in optimal cutting temperature (OCT) at the time of sectioning. The tissue was thick sectioned (50 µm, free floating) in a cryostat (Microm, HM 505 E) in a series of 10 vials containing 1xPBS. Samples were stored in PBS with 0.2% sodium azide at 4 °C, and protected from light exposure.

Prior to antibody incubation, sections were blocked with 10% serum in PBS with 0.2% triton × (Sigma Aldrich) and 0.1% tween 20 (Sigma Aldrich), except for anti-Gad67 staining which did not include detergents. Primary antibodies (Table 1) were detected with Alexa conjugated secondary antibodies (Jackson labs and Abcam) or AMCA conjugated secondary antibodies (Vector) in 5% serum. VECTASHIELD DAPI was used for double staining and VECTASHIELD without DAPI was used for triple staining when mounting sectioned tissue onto polyprep slides.

Table 1 Antibodies used for immunofluorescence detection.

Antigen	Raised in	Dilution	Source	Marker of	
NeuN	Mouse	1:125	Millipore	Neuron	
GFAP	Rabbit	1:500	Dako Cytomation	Astrocyte	
GFP	Chicken	1:250	Abcam Inc.	GFP	
Sox2	Rabbit	1:250	Millipore	Neuronal precursor	
Olig2	Mouse	1:1000	Millipore	Oligodendrocyte	
DCX	Mouse	1:250	Abcam Inc.	Neuroblast	
GAD67	Mouse	1:1000	Millipore	Inhibitory neuron	
Calretinin	Rabbit	1:1000	Millipore	Inhibitory neuron	
Somatostatin	Rat	1:200	Millipore	Inhibitory neuron	
BLBP	Rabbit	1:1000	Abcam Inc.	Stem cells	

Cell Counting

Unbiased estimated counts of astrocytes expressing glial fibrillary acidic protein (GFAP), neurons expressing NeuN, and oligodendrocytes and oligodendrocyte precursors expressing Olig2 were obtained using StereoInvestigator software (Microbrightfield, MBF) coupled with an AxioCam MRm on the Zeiss axioimager microscope equipped with an ApoTome.2. The Optical Fractionator probe on Stereoinvestigator was employed, with tops of cells counted in three-dimensional counting boxes, which were set to specific parameters (Table 2). For counts of the cortex and hippocampus, 50 µm sections were sampled every 20th (cortex) and every 10th section (Hippocampus). The CA1, CA2, and CA3 regions were counted together and included the stratum lacunosum, stratum radiatum, stratum pyramidale and the stratum oriens. The DG was counted separately. For the cortex, at these ages, the expression of GFP seemed fairly ubiquitous and so the cortex was counted in its entirety using stereology as outlined above and in Table 2. Fluorescence images were acquired through StereoInvestigator imaging software. Of note, cell counts indicated that presence of the GFP transgene did not cause cell lethality in cortical or hippocampal cells. For P7 counts, 3 animals per genotype were counted for NeuN, GFAP, and Olig2 (same animals for all 3 counts). For 1 month old animals, 3 animals per group were counted for cortex and CA, and 4 animals in the GFP+ group for the DG. The GFAP/NeuN/GFP counts were determined from triple stained samples. The Olig2 counts were from 3 animals per group.

Table 2 Parameters used for stereological cell counts.

Time point	Area	Counting frame (µm)	Grid size (µm)	Dissector height (µm)	Guard zone height (µm)	Section thickness (µm)	
P7	Cortex	75 × 75	650 × 700	15	1	50	
	DG	75 × 75	250 × 250	15	1	50	
	CA	75 × 75	500 × 500	15	1	50	
1 month	Cortex	75 × 75	1,000 × 1,000	15	1	50	
	DG	75 × 75	275 × 200	15	1	50	
	CA	75 × 75	350 × 250	15	1	50	

To determine the percentage of GFP+ cells that were neuronal stem cells expressing Sox2 and neuroblasts expressing DCX (3 animals per group), we acquired z stack images of the anterior DG hemisphere and a section 500 µm posterior to this first section. The z stack images obtained to count SOX2 markers were 19 µm thick with each slice 1 µm thick from 3 GFP+ mice and 2 for the control mice). The z stack images acquired to count DCX markers were 10 µm thick with each slice 1 µm thick (3 animals per group).

Statistical analysis

Data from the StereoInvestigator software were entered into Excel, imported to JMP Pro 11, and analyzed by student t-tests, or ANOVA, using SAS.

Results

Fgfr1 is expressed in various cells types of the dentate gyrus, CA, and subventricular zone (SVZ) at P7

Previous in situ studies indicated high levels of Fgfr1 mRNA in the hippocampus, including DG and CA regions. To investigate the cell types expressing Fgfr1 in the hippocampus of P7 control and tgfgfr1-EGFP+ mice, samples were stained by immunocytochemistry for cell-type markers to identify each of the following: GFAP+ stem cells and astrocytes, NeuN+ neurons, BLBP+ stem cells, and DCX+ neuroblasts. We also used an anti GFP antibody to ensure detection of GFP in fixed tissue. However, Gfp expression is visible without amplification, especially in the hippocampus and subventricular zone. The cell-type marker staining (red fluorescence) was imaged alongside GFP immunostaining (green fluorescence) to reveal which cell types express Fgfr1 and with DAPI counterstaining (blue fluorescence) to image nuclei (Fig. 1). The colocalization of a cell type marker and Fgfr1 expression is indicated by yellow fluorescence caused by overlapping red and green fluorescence. GFAP and GFP immunostaining showed GFP+ cells in the GFAP+ stem cells of the subgranular zone (SGZ) of the DG of tgfgfr1-EGFP+ mice (Figs. 1A–1C, Fig. S1, and https://figshare.com/s/36bc336d5b0ccb95efe8) with little to no green fluorescence in GFP-controls (Figs. 1D–1F). In the CA region of the hippocampus, GFAP+/GFP+ cells were primarily observed surrounding the stratum pyramidale, in the stratum radiatum, stratum oriens, as well as the white matter above the CA region (Figs. 1G–1I, Fig. S1) with little to no green fluorescence in GFP-littermate controls (Figs. 1J–1L). There was strong GFP fluorescence in the stratum pyramidale of the CA region. This GFP staining colocalized with NeuN+ cells (neurons) (Figs. 2A–2C, Fig. S2) with little to no green fluorescence in GFP-littermate controls (Figs. 2D–2F). NeuN+/GFP+ cells were also observed in some, but not all, granule cell layer neurons of the developing DG of tgfgfr1-EGFP+ mice (Figs. 2G–2I) with little to no green fluorescence in GFP-controls (Figs. 2J–2L). Stereological analysis of tgfgfr1-EGFP expression in GFAP+ cells of the DG showed that of GFP+ cells, 24% ± 8% are GFAP positive, and of GFAP+ cells, 50% ± 8% are GFP+. Of GFP+ cells in the CA, 50% ± 5% are GFAP positive and of GFAP+ cells in the CA, 43% ± 9% are GFP positive (Table 3). One-way ANOVA statistical analysis revealed that the total number of GFAP+ cells in the DG and CA of tgfgfr1-EGFP+ are not significantly different from their littermate controls (Table 3).

Figure 1 Fgfr1 expression in GFAP+ cells of the hippocampus at P7.

GFAP (A, D) GFP (B, E) immunostaining of the DG in P7 tgfgfr1-EGFP+ mice (A–C, n = 3) and tgfgfr1-EGFP-littermate controls (D–F, n = 3) demonstrated strong GFAP/GFP colocalization in cells of the SGZ (arrowheads) and their radial fibers into the GCL (small arrows). GFAP and GFP immunostaining of the CA region in tgfgfr1-EGFP+ mice (G–I, n = 3) and tgfgfr1-EGFP-controls (J–L, n = 3). GFP+ staining was observed in stratum pyramidale (SP) as well as in cells above (stratum oriens, SO) and below (stratum radiatum, SR) this layer (H). GFAP+/GFP+ colabeling (arrowheads) was observed in primarily in the SO and SR within the CA region, and in the white matter dorsal to the hippocampus. Scale bar is 25 µm. SGZ, subgranular zone; GCL, granule cell layer; SP, stratum pyramidale.

Figure 2 Fgfr1 expression in NeuN+ cells of the hippocampus at P7.

NeuN (A, D) and GFP (B, E) immunostaining of the CA region in tgfgfr1-EGFP+ mice (A–C, n = 3) and tgfgfr1-EGFP-controls (D–F, n = 3). Neun+/GFP+ staining was observed in stratum pyramidale (SP) of the CA region. NeuN (G, J) and GFP (H, K) immunostaining of the DG in tgfgfr1-EGFP+ mice (G–I, n = 3) and tgfgfr1-EGFP-controls (J–L, n = 3). There was some colabeling in the granule cell layer between GFP+ and NeuN+ cells. Scale bar is 25 µm in A–D and 50 µm in G–L.

Table 3 Stereology Results for Hippocampal DG, CA, and Cortex.

Age	Area	Marker	# of Marker+ cells	P-value	# of GFP+ cells	% of GFP+ cells expressing Marker	% of Marker+ cells expressing GFP	
P7	DG	GFAP	5614 ± 467	0.37	15,974 ± 7447	24% ± 8%	50% ± 8%	
	CA	GFAP	51,258 ± 1741	0.42	46,413 ± 14,851	50% ± 5%	43% ± 9%	
	Cortex	GFAP	41,2147 ± 22,494	0.25	12,20518 ± 53,849	32% ± 4%	93% ± 2%	
1 Month	DG	NeuN	75,5542 ± 22,8698	0.52	24,3542 ± 12,945	16% ± 6%	5% ± 0.7%	
		GFAP	14,9161 ± 22,540	0.53	24,3542 ± 12,945	51% ± 3%	85% ± 4%	
	CA	NeuN	67,0709 ± 59,607	0.27	37,9768 ± 27,634	25% ± 7%	15% ± 6%	
		GFAP	28,6614 ± 39,645	0.86	37,9768 ± 27,634	61% ± 9%	81% ± 3%	
	Cortex	GFAP	17,77872 ± 87,6236	0.66	23,21696 ± 10,56829	57% ± 7%	83% ± 6%	
		NeuN	49,62422 ± 22,64757	0.97	23,21696 ± 10,56829	25% ± 4%	12% ± 2%	
		Olig2	80,0755 ± 13,7359	0.85	15,74962 ± 19,5563	29% ± 0.9%	58% ± 0.08%	

We further explored Fgfr1 promoter driven expression of Gfp in the postnatal neurogenic niches of the DG and SVZ of the lateral ventricles at P7. Immunostaining for DCX (neuroblasts) and GFP extensively colocalized in the SGZ and granule cell layer (GCL) of the DG (Figs. 3A and 3B, low and high magnification respectively) with nearly all DCX positive cells co-expressing GFP. Postnatally, the SVZ of the lateral ventricles gives rise to olfactory bulb neurons, astrocytes and oligodendrocytes (Chaker, Codega & Doetsch, 2016). The kind of olfactory bulb neuron generated depends on the location of the stem cell within the VZ (Fiorelli et al., 2015; Chaker, Codega & Doetsch, 2016). GFP+ cells were observed in the SVZ, consistent with previous studies of postnatal Fgfr1 expression (Fagel et al., 2009). However, DCX did not colocalize with GFP staining in the SVZ of the lateral ventricles (Fig. 3C), in contrast to what was observed in the DG of the hippocampus. This difference is likely reflective of the differences in cell type generated, granule cell neurons that are projection neurons in the SGZ and largely interneurons in the SVZ. Immunostaining with BLBP (stem cells) and GFP antibodies revealed colocalization of Fgfr1 driven GFP in the subgranular zone (SGZ) of the DG (Figs. 3D and 3E, low and high magnification, respectively) as well as BLBP+ cells of the SVZ (Figs. 3F and 3G).

Figure 3 Fgfr1 expression in neuroblasts of the hippocampus, and stem cells of the hippocampus and SVZ at P7.

DCX and GFP colocalize in neuroblasts of the P7 mice in the granule cell layer of the DG in the hippocampus (A, low magnification, and B, high magnification), but not in the SVZ of the lateral ventricles (C). Many of the BLBP+ cells colocalize with GFP in the stem cells of the SGZ of the DG (D, low magnification, and E, high magnification), as well as stem cells in the SVZ (F, G). Scale bar is 50 µm in A, C, D, F and 25 µm in B and E, and 12.5 µm in G. Arrows indicate examples of double labeled cells, dashed lines denote SVZ.

Fgfr1 is expressed in cortical GFAP+ astrocytes and NeuN+ neurons at P7

To determine which cortical cells express Fgfr1, we stained the cortical tissue of P7 tgfgfr1-EGFP+ and control mice with GFAP and NeuN antibodies. Whereas GFP+ cells colocalized with GFAP+ astrocytes throughout the cortex (Figs. 4A–4C), GFP+ cells colocalized with NeuN+ cells (neurons) mostly in layers 5 and 6 of the cortex at this age (Figs. 4D–4F). Stereological analysis of the cortex region revealed that of GFP+ cells in the cortex, 32% ± 4% are GFAP positive. Of the cortical GFAP+ cells, 93% ± 2% are GFP positive. This analysis indicates that most Gfap expressing astrocytes also express Fgfr1. The total number of cortical GFAP+ cells of tgfgfr1-EGFP+ mice was not significantly different from the number of cells in littermate controls (Table 3), suggesting that the insertion of the transgene does not significantly alter the number of astrocytes expressing Gfap.

Figure 4 Fgfr1 expression in cortex of P7 mice.

Cortical GFAP, NeuN, BLBP and GFP immunostaining in P7 tgfgfr1-EGFP+ mice (A, B, D, E, G–J, n = 3) and tgfgfr1-EGFP-controls (C, F, n = 3). GFP+ cells colocalize with GFAP+ cells (A, B). GFP+ cells weakly colocalize with NeuN+ at layers 5 and 6 of the cortex (D, E, dashed line separates cortical plate and subcortical white matter). GFP+ cells colocalize with Olig2+ cells in the cortex (G–I) and some of the OLIG2+ cells of the subcortical white matter and SVZ (J). GFP+ cells colocalize with BLBP+ cells throughout the layers of the cortex (K–M) as well as the BLBP+ cells and Bergmann glia of the cerebellum (N). Scale bar is 50 µm. 1, 2, 3, 4, 5, 6, layers of the cortex. Arrows indicate examples of double labeled cells.

We examined whether tgfgfr1-EGFP+ was expressed in oligodendrocytes and their precursors by staining for OLIG2 in the cortex. We found that OLIG2+ staining colocalized with GFP in the cortex (Figs. 4G–4I). We also found sparse colabeling of OLIG2+ cells in the SVZ and subcortical white matter (Fig. 4J). GFP expression within these cells appears to be lower than that of surrounding cells, and may reflect residual GFP expression once progenitor cells that express Fgfr1 have divided and an oligodendrocyte has initiated differentiation to a more mature subtype. At P7, radial glia of the cortex are undergoing soma translocation to become astrocytes, and can be detected with BLBP. BLBP and GFP immunostaining revealed strong colocalization throughout the cortex (Figs. 4K–4M). At P7, GFP staining colocalized with BLBP+ stem cells and Bergmann Glia in the developing cerebellum (Fig. 4N). This finding is consistent with the participation of FGFR1/FGFR2 signaling in Bergmann Glia morphology and cerebellar development (Müller Smith et al., 2012). Furthermore, GFP+ cells colocalized with GFAP+ astrocytes of the cerebellum, but not within granule cell layer or molecular layer neurons that stain with NeuN. GFP+ cells likely colocalized with NeuN-Purkinje neurons based on the location and size of GFP+ cells in the Purkinje layer (Fig. S3 and https://figshare.com/s/722ecee87a3f32427759). While counts were not obtained from regions such as the striatum or the thalamus, we observed GFAP colocalization with GFP in multiple areas of the brain, and this seemed to be a general feature of GFAP positive cells.

Fgfr1 is expressed in specific cell types of the hippocampus and SVZ at 1 month

We next investigated Fgfr1 expression in 1-month tgfgfr1-EGFP+ mice. Immunostaining for SOX2+ cells (stem cells), DCX+ cells (neuroblasts), NeuN+ neurons, GFAP+ astrocytes and stem cells, and OLIG2+ oligodendrocytes was performed in combination with GFP immunostaining to determine which cell types express Fgfr1 in the DG and CA of the hippocampus, as well as the SVZ. GFP+ cells colocalized with SOX2+ cells (Figs. 5A–5E, Figs. S4A and S4B) and DCX+ cells (Figs. 5F–5J, Figs. S4C and S4D) in the SGZ and granule cell layers. To determine the percentage of GFP+ cells that were SOX-2 positive and DCX positive, we obtained z stack images of an anterior dentate gyrus hemisphere and posterior dentate gyrus hemisphere, and performed counts from these images. Of the GFP+ cells counted in the z stack, 33% ± 2% are SOX2 positive. Conversely, the majority of SOX2+ cells, 71% ± 2%, are GFP positive (Fig. 5E). Of the GFP+ cells counted in the z stack, 64% ± 0.9% are DCX positive and of DCX+ cells, 86% ± 2% are GFP positive (Fig. 5J). Triple staining with GFP, GFAP and NeuN antibodies revealed that GFAP+ cells (stem cells) and NeuN+ cells (neurons) colocalized with GFP+ cells in the SGZ (Figs. 5K–5N, Fig. S4E) and in the CA region (Figs. 5O–5R and Fig. S4F), respectively. GFP+ cells did not colocalize with OLIG2+ cells (oligodendrocytes) in the DG (Fig. S5). Taken together, these results indicate that GFP (Fgfr1) was expressed in stem cells and neuroblasts of the DG. Stereological analysis of tgfgfr1-EGFP expression in NeuN+ cells of the DG showed that of GFP+ cells, 16% ± 6% are NeuN positive and of NeuN+ cells, 5% ± 0.7% are GFP positive. Of GFP+ cells in the CA, 25% ± 7% are NeuN positive and of NeuN+ cells, 15% ± 6% are GFP positive. Stereological analysis of tgfgfr1-EGFP expression in GFAP+ cells of the DG showed that of GFP+ cells, 51% ± 3% are GFAP positive and of GFAP+ cells, 85% ± 4% are GFP positive. Of GFP+ cells in the CA, 61% ± 9% are GFAP positive and of GFAP+ cells, 81% ± 3% are GFP positive. These findings indicate that most GFAP positive astrocytes express Fgfr1 at 1 month. The total number of SOX2+ cells, DCX+ cells, NeuN+ cells and GFAP+ cells of tgfgfr1-EGFP+ mice were not significantly different from their littermate controls in the DG and CA (Sox2: p = 0.23, DCX: p = 0.88 and Table 3).

Figure 5 Fgfr1 expression in the dentate gyrus and CA of 1-month mice.

A majority of SOX2+ cells (A, C) in the DG of tgfgfr1-EGFP+ mice colocalize with GFP (B, C, arrows indicate examples of double labeled cells) compared to GFP- controls (D). Counts of SOX2+ and GFP+ cells were made directly from Z stack images (19 µm thick, 1 µm steps, from Z stack images using StereoInvestigator software from MBF). The relative proportions of single, and double-labeled cells in the DG were quantified (E), with 71% of SOX2 positive cells expressing GFP. A majority of DCX+ (F, H) cells in the DG also express GFP (G, H, arrows indicate examples of double labeled cells) compared to GFP-controls (I). Counts for DCX+ and GFP+ were performed on 10 µm thick, 1 µM step size, Z stack images using Image J (n = 3) of the dentate gyrus. The relative proportions of single, and double-labeled cells in the DG were quantified (J) with 85% of DCX+ cells expressing GFP. GFAP (Red), NeuN (Blue), and GFP triple immunolabeling was perfused on 1-month old mice. We observed GFAP and GFP colabeling in both the SGZ of the DG (K–N, arrows), as well as in the CA region (O–R) where most GFAP+ cells are present in the stratum oriens or stratum radiatum. NeuN+/GFP+ double positive cells were primarily observed in the stratum pyramidale (SP) of the CA region in the hippocampus (double arrows). These cells were quantified by unbiased stereology with the StereoInvestigator (Table 3). Scale bar in I is 25 µm for images A–D and F–I, Scale bar in R is 50 µm for images K–R. Arrows indicate examples of double labeled cells.

Many GFAP+ cells also express GFP in the SVZ (Figs. 6A–6H) of 1 month tgfgfr1-EGFP+ mice. Colocalized GFP positive cells and GFAP positive fibers were observed both within the SVZ and in the white matter above it (Figs. 6D, 6G and 6H). There was also significant colocalization of GFP with SOX2+ cells (Figs. 6I–6L) indicating that GFAP+ and SOX2+ stem cells of the SVZ express Fgfr1. Similar to what was found at P7, GFP+ cells did not colocalize with DCX+ cells (neuroblasts) in the SVZ (Figs. 6M–6O) or in the DCX+ cells of rostral migratory stream (RMS) as it enters the olfactory bulb (Fig. 6P). In the rostral migratory stream, GFP staining surrounds the DCX+ cells as would be expected from astrocytes surrounding the migrating neuroblasts.

Figure 6 Fgfr1 expression in the SVZ and rostral migratory stream of 1-month mice.

GFP (A, E, D, H), GFAP (B, F, G, H), NeuN (C, D), immunostaining of the SVZ in 1-month tgfgfr1-EGFP+ mice (A–H, n = 3). GFP+ cells of the SVZ colocalized with GFAP+ cells (D, and G, H. Arrowheads in E–H = GFAP/GFP+ cells and GFAP+ fibers. GFP (I, K, L) and SOX2 (J, K, L) staining demonstrated that many, but not all SOX2+ cells also colabel with GFP+ (K and L, arrows indicate double labeled cells, DAPI staining included in L). In contrast, DCX+ neuroblasts in the SVZ and rostral migratory stream did not colabel with GFP (M–P). Scale bar is 50 µm in A–D and M–P and 25 µm in E–L. Dashed lines indicated SVZ region examined.

Fgfr1 is expressed in cortical GFAP+ astrocytes, NeuN+ neurons and OLIG2+ cells at 1 month

To determine which cortical cells express Fgfr1, we examined cortical tissue of 1-month old tgfgfr1-EGFP+ and control mice immuonostained with GFAP, NeuN or OLIG2 antibodies alongside GFP antibodies. Analysis of Fgfr1 expression in the cortex revealed that GFP+ cells colocalized with GFAP+ and NeuN+ cells (Figs. 7A–7E, compared to control 7F). Of note, GFP immunofluorescence appeared less brightly fluorescent in cells colocalizing with NeuN compared to those colocalizing with GFAP. GFP+ cells also colocalized with OLIG2+ cells (Figs. 7G–7H) with little to no green fluorescence occurring in littermate controls (Fig. 7I). Stereological analysis of the cortex revealed that of GFP+ cells, 57% ± 7% are GFAP positive. Of the GFAP+ astrocytes, 83% ± 6% are GFP positive. Of GFP+ cells, 25% ± 4% are NeuN positive and of NeuN+ cells, 12% ± 2% are GFP positive. Of GFP+ cells, 29% ± 0.9% are OLIG2 positive and of OLIG2+ cells, 58% ± 0.08% are GFP positive. The total number of cortical GFAP+, NeuN+, and OLIG2+ cells of tgfgfr1-EGFP+ mice were not significantly different from their littermate controls (Table 3).

Figure 7 Fgfr1 expression in the cortex of 1-month mice.

Cortical GFAP (Red), NeuN (Blue), and GFP immunostaining in 1-month tgfgfr1-EGFP+ mice (A–E, n = 3) and control tgfgfr1-EGF-mice (F, n = 3). GFP+ cells colocalized with GFAP+ (arrows) (A, B, merged in D, low magnification in E), and some NeuN+ (double arrows) (B, C, merged in D, lower magnification in E). Immunostaining for Olig2+ in tgfgfr1-EGFP+ mice (G, H, Red, Olig2; Blue, DAPI) compared to tgfgfr1-EGF-mice (I) demonstrated that many Olig2+ cells colocalized with GFP. DAPI staining omitted from low magnification pictures in G and I. Scale bars are 50 µm. Dashed line denotes top of cortex.

Fgfr1 is expressed in various cell types of the hypothalamus at P7 and 1 month

Comparing Fgfr1 expression in different regions of the P7 and 1-month old mice, we observed GFP expression in cells of the hypothalamus and among cells lining the third ventricle, within the arcuate nucleus and median eminence (Fig. 8, Fig. S6 and https://figshare.com/s/f17d6889939bd680152d). Compared to the control hypothalamus (Figs. 8A–8C), GFP+ cells strongly colocalized with SOX2+ cells throughout the hypothalamus and third ventricle and arcuate nucleus (Figs. 8D–8I) in 1-month tgfgfr1-EGFP+ mice. GFAP+ tanycytes, including those of the arcuate nucleus were among the hypothalamic cells observed to express Fgfr1 both at 1 month and at P7 (Figs. 8J–8L). Tanycytes participate in neuroendocrine regulation and transport of molecules from the CSF to the hypothalamus, release of gonadotropin-releasing hormone, and production of triiodothyronine (T3) in the brain (Rodriguez et al., 2005). GFP expression was not observed in NeuN+ or OLIG2+ cells of the hypothalamus (Figs. 8M–8O, Figs. S6A–S6D). At P7, we also observed colocalization of BLBP and GFP in the hypothalamus near the third ventricle, but not many in the median eminence (Figs. S6E and S6F, https://figshare.com/s/f17d6889939bd680152d).

Figure 8 Fgfr1 expression in the hypothalamus.

SOX2+ positive cells were observed along the third ventricle of tgfgfr1-EGF-control mice (A–C) and tgfgfr1-EGF+ mice (D–I). These SOX2+ cells also express GFP under the Fgfr1 promoter (D–I). GFAP+ tanacytes along the third ventricle and arcuate nucleus also colocalize with GFP, as do GFAP+ cells within the hypothalamus (J–L). NeuN+ cells in the hypothalamus do not colocalize with GFP (M–O). A–L, 1 month; M–O, P7. Scale bar is 50 µm.

Calretinin (CR) and somatostatin (SST) neurons express Fgfr1 in one-month old mice

The NeuN+/GFP+ neurons observed at 1 month were not restricted to any specific cortical layer, and a minority population of GFP colocalized with NeuN+ neurons. We therefore sought to determine if inhibitory neurons express Fgfr1. Tissue from one-month old tgfgfr1-EGFP+ mice (Figs. 9B–9C) and their control littermates (Fig. 9A) was stained for GAD67. Some GAD67+ cells colocalized with GFP in the cortex (Fig. 9B), but not in the hippocampus (Fig. 9C) or SVZ (Fig. 9D). Our previous investigations determined that GFP was not colocalized with PV+ interneurons (Smith et al., 2014). This led to the experiments in which staining the one-month old tissue with calretinin (CR) and somatostatin (SST) was performed alongside GFP immunostaining. CR+ inhibitory neurons express GFP in the cortex (Figs. 9E and 9F), DG (Fig. 9G), and SVZ (Fig. 9H). Some SST+ inhibitory neurons express GFP in the anterior cingulate of the cortex (Figs. 9I and 9J), but none were observed in the DG (Fig. 9K).

Figure 9 Fgfr1 expression in the 1-month interneurons.

GAD67, CR, SST, and GFP immunostaining of the cortex, DG, and SVZ in one-month old tgfgfr1-EGFP-controls (A, E, I, n = 2 SST, n = 3 GAD67, CR) and tgfgfr1-EGFP+ mice (B–D, F–H, J, K, n = 2 SST, n = 3 GAD67, CR). GFP+ cells colocalized with GAD67+ cells in the cortex (B, arrows), but not in the DG (C) and SVZ (D). Some CR+ cells express GFP in the cortex (F), DG (G) and SVZ (H). Some SST+ cells express GFP in the anterior cingulate of the cortex (J), but not in the DG (K). Scale bars are 50 µm. Dashed lines indicate SVZ, AC, anterior cingulate. Arrows denote double labeled cells.

Discussion

Through immunostaining for GFP in tgFgfr1-EGFP+ mice of the tgFGFR1-EGFPGP338Gsat line, we examined the cell specific expression of Fgfr1 in the cortex, hippocampus, SVZ, hypothalamus, and cerebellum from P7 to one-month of age. In postnatal brains, Fgfr1 is highly expressed in the hippocampus, SVZ, hypothalamus, and in numerous cells of the cortex. In the hippocampus, Fgfr1 was expressed in DCX+ neuroblasts, in GFAP+ stem cells and astrocytes, and Sox2+ stem cells of the SGZ, and in neurons of the CA. Furthermore, Fgfr1 was expressed in astrocytes, neurons, and oligodendrocyte lineages. Fgfr1 expression in cortical neurons was not restricted to a single layer and colocalized with CR+ and SST+ inhibitory neurons. Fgfr1 expression was also observed in Bergmann glia and GFAP+ astrocytes of the cerebellum.

The tgFgfr1-EGFP+ line has been shown to be a robust model in identifying which cell types express Fgfr1, corroborating the findings of previous in situ hybridization studies (El-Husseini, Paterson & Shiu, 1994; Gonzalez et al., 1995; Belluardo et al., 1997; Bansal et al., 2003; Ohkubo et al., 2004) and more modern molecular profiling studies (Lovatt et al., 2007; Cahoy et al., 2008; Doyle et al., 2008; Azim, Raineteau & Butt, 2012). This transgenic line gives the added benefit of allowing simple identification of the subtype of cells expressing Fgfr1 by immunofluorescence staining, since GFP is regulated by the Fgfr1 promoter. Unlike the use of antibodies for FGFR1, which may have cross reactivity with similar FGFR family members, we can have greater confidence that we are identifying Fgfr1 expression accurately within a cell.

Previous studies have shown that inactivation of FGFR1/FGFR2 and FGFR1/FGFR2/ FGFR3 signaling in the developing dorsal VZ results in premature depletion of radial glial stem cells and a smaller cortex, whereas loss of Fgfr1 alone is sufficient to result in reduced hippocampal volume and reduced hippocampal stem cell proliferation during embryogenesis and the early postnatal period (Ohkubo et al., 2004; Kang et al., 2009; Stevens et al., 2010b; Rash et al., 2011). Our studies of Fgfr1 expression in embryos and at P1 will be described elsewhere, but support the role of FGFR1 in telencephalic development. At P7, Fgfr1 was expressed in BLBP positive cells. At this age, the BLBP+ radial glial cells are undergoing soma translocation and differentiating into astrocytes of the cortical parenchyma. FGF signaling may be participating in the soma translocation process, as previously described for the glial cells of the indusium griseum (Smith et al., 2006), or it may be a general factor expressed by astrocytes since a majority of GFAP+ cells expressed GFP. It is interesting to note that as animals matured from P7 to one-month of age, the relative amount of GFAP/GFP colocalization was increased. This may reflect a maturation of astrocytic glia from BLBP expressing radial glia stem cells undergoing gliogenesis and soma translocation to mature astrocytes that express higher levels of GFAP. BLBP is also expressed in Bergmann glia of the cerebellum, which are cells that have dual roles as stem cells and as a scaffold for granule cell neuron migration and alignment of Purkinje neuron dendrites. The finding that Fgfr1 is expressed within these cells is consistent with the demonstrated occurrence of cooperative signaling between FGFR1 and FGFR2 in the formation of the cerebellum, and specifically, in the morphology and pial attachment of Bergmann glia of the cerebellar anlage (Lin et al., 2009; Müller Smith et al., 2012). FGFR1 may be playing dual roles in these cells as a factor that supports their proliferation, as well as structural morphology, potentially by interactions with cell adhesion molecules (Williams et al., 1994; Polanska, Fernig & Kinnunen, 2009).

At P7, and one month, Fgfr1 was expressed in the DG and CA regions of the hippocampus, as well as in the SVZ, hypothalamus, and all layers of the cortex. In the DG, expression is primarily found in stem cells as well as DCX positive cells. The Fgfr1 expressing cells in the CA region at P7 and 1 month colocalize with GFAP positive astrocytes and NeuN positive neurons in the stratum pyramidale. Fgfr1 expression within GFAP+, and SOX2+ positive cells of the SGZ in the DG indicates that FGFR signaling is contributing to stem cell maintenance and proliferation in the DG, and that alterations in FGFR signaling may contribute to depletion of DG stem cells (Ohkubo et al., 2004). As reviewed by (Patricio et al., 2013), reduced hippocampal proliferation has been observed in animal models of depression, and antidepressant therapies all promote either hippocampal proliferation or the survival and integration of newborn neurons (Patricio et al., 2013). FGFR signaling also participates in development of hippocampal synaptogenesis (Cambon et al., 2004; Umemori et al., 2004; Terauchi et al., 2010). Activation of FGFR1 promotes synapse formation in hippocampal neurons (Cambon et al., 2004; Terauchi et al., 2010; Dabrowski et al., 2015). The expression of Fgfr1 in DCX positive cells of the hippocampus, as well as neurons in the CA region supports the evidence that FGFR1 plays additional roles in synapse formation and integration into the hippocampal circuitry. These findings have implications for hippocampal function in HPA axis regulation. Changes in Fgfr1 expression, and FGF2 levels in the hippocampus are linked to major depression and anxiety, as well as to responses to antidepressants (Evans et al., 2004; Newton & Duman, 2004; Gaughran et al., 2006; Turner et al., 2008; Elsayed et al., 2012). It is hypothesized that Fgf2 and Fgfr1 are downregulated in depression and anxiety, and this downregulation can be partially reversed by antidepressant treatment (Turner, Watson & Akil, 2012b; Turner, Watson & Akil, 2012a). The chronic unpredictable stress model causes anhedonia and is a model of depression that results in decreased (mRNA) Fgfr1 levels in the prefrontal cortex (Elsayed et al., 2012). Antidepressant treatment resulted in increased (mRNA) Fgf2 levels, and the antidepressant effects could be blocked by an FGFR1 inhibitor. These authors further showed that FGF2 administration also had an antidepressant effect (Elsayed et al., 2012). The tgfgfr1-EGFP+ mouse line can be utilized to further examine the effects of stress and depression models on Fgfr1 expression.

GFAP positive astrocytes and BLBP cells in the SVZ at P7 are expressing GFP, indicating that stem cells of the SVZ express Fgfr1. GFAP positive astrocytes at 1 month also express Fgfr1. Our findings in the SVZ are consistent with studies of single cell transcriptomics of the SVZ (Azim, Raineteau & Butt, 2012; Azim et al., 2015; Llorens-Bobadilla et al., 2015). For example, molecular studies have identified that CD133+ stem cells of the pallium, capable of generating neurons or glia, have higher levels of Fgfr1 mRNA compared to other SVZ cell types of the lateral SVZ (Azim, Raineteau & Butt, 2012). Llorens-Bobadilla et al. (2015), identified subpopulations of cells within the SVZ via principle component analysis of co-expressed gene transcripts from single cells, and identified that Fgfr1 was enriched in Neural Stem Cells (NSCs) that express glial/NSC markers such as Fabp7 (BLBP), Slc1a3 (GLAST), Hes5, and Aldh1l1. Fgfr1 was co-expressed with Category III cells in that study, which included dynamically regulated genes that change expression as NSC go from a quiescent state to an activated state. Future studies could employ tgfgfr1-EGFP+ mice to explore the expression of Fgfr1 during remyelination and response to injury. Fgfr1 expressing cells were also colocalized with SOX2 positive cells at 1 month. At P7 and 1 month, Fgfr1 expressing cells in the SVZ do not colocalize with DCX positive neuroblasts and NeuN positive neurons, as confirmed by single cell transcriptomics studies. Thus, DCX positive cells express Fgfr1 in one of the postnatal neurogenic regions—the DG. The difference in Fgfr1 expression observed between DCX positive cells of the DG and SVZ implies that the staining in DCX positive cells of the hippocampus is not due to residual GFP protein that has not been degraded once a daughter cell born from a stem cell stops expressing Fgfr1, suggesting we can further accurately identify Fgfr1 expression within cells using this transgenic model. An alternative explanation is that the lack of DCX in the neurons born from stem cells in the SVZ is that they have a completely different profile and fate (olfactory bulb neurons) compared to the newborn granule neurons of the hippocampus, and thus, would not be expected to express Fgfr1 in the same way.

Most GFAP positive astrocytes throughout the cortex coexpressed Fgfr1 at similar levels, and the majority of the Fgfr1 expressing cells were GFAP+ astrocytes. Fgfr1 expression seemed to be a general feature of GFAP positive cells and glial stem cells throughout development. A minority of Fgfr1 expressing cells were NeuN positive neurons at P7 and one month. While it was previously believed that Fgfr2 is expressed in glia and that Fgfr1 is mostly expressed by neuronal populations in the adult cortex, our findings support the view that most of the cortical cells expressing Fgfr1 in the adult brain are, in fact, astrocytes or oligodendrocytes (Turner, Watson & Akil, 2012a). In this study we see that most GFAP positive astrocytes express Fgfr1 and a small proportion of NeuN positive neurons express Fgfr1 in one-month old mice. A higher percentage of Fgfr1 expressing cells are GFAP positive astrocytes rather than NeuN positive neurons. Thus, astrocytes constitute a majority of Fgfr1 expressing cells in the adult brain. This idea is consistent with previous mRNA profiling of astrocytes in which Fgfr1 was identified as a gene with enriched expression in astrocytes as compared to other cell types in the brain (Lovatt et al., 2007; Cahoy et al., 2008; Doyle et al., 2008).

An imbalance of excitatory and inhibitory neurons, along with hyperactivity has been documented in certain neurological disorders (Benes et al., 2000; Volk et al., 2000; Kalanithi et al., 2005; Akbarian & Huang, 2006; Hashimoto et al., 2008; Kataoka et al., 2010; Gonzalez-Burgos, Fish & Lewis, 2011; Volk & Lewis, 2013). Inactivation of Fgfr1 leads to a decrease in the abundance of parvalbumin interneurons in the cortex and is associated with an increase in hyperactivity (Muller Smith et al., 2008). A subsequent study demonstrated that a decrease in interneurons occurs postnatally and that Fgfr1 expression does not occur in parvalbumin expressing interneurons (Smith et al., 2014). Astrocytes lacking Fgfr1 were less capable of supporting the maturation of Gad67 expressing cells than control astrocytes. Here, we find that Fgfr1 is expressed by a majority of astrocytes in the cortex and hippocampus throughout the maturational period of parvalbumin positive interneurons, further supporting the importance for FGFR1 signaling in astrocytes. Current studies are seeking to explore how FGFR1 signaling within astrocytes contributes to interneuron maturation.

Here, we tested other interneuron markers, including GAD67, calretinin, and somatostatin. At one month, GAD67, a marker for inhibitory neurons, colocalizes with Fgfr1 expressing cells in the cortex and not in the DG. Calretinin interneurons express Fgfr1 throughout the cortex, and in the DG, whereas somatostatin interneurons express Fgfr1 at the anterior cingulate of the cortex. Brandt et al. (2003) demonstrated newborn calretinin positive cells in the DG do not express GABA, offering a probable explanation as to why GAD67 is not observed colocalizing with Fgfr1 expressing cells in the DG.

The hypothalamus, responsible for controlling hormonal production, stress regulation, and feeding behaviors, has been found to contain a neural stem/progenitor cell niche (Robins et al., 2013). Our results show that GFAP positive astrocytes are Fgfr1 expressing cells. The morphology and location of the Fgfr1 expressing cells are consistent with the cells being tanycytes of the hypothalamus (Robins et al., 2013). The median eminence of the hypothalamus contains Fgfr1 expressing cells that also express SOX2. Fgfr1 expressing cells that were not identified are similar in morphology and location to the neurons that release gonadotropin-releasing hormone (GNRH1). These results are in congruence with a study, which demonstrated FGFR1’s role in the targeting of GNRH1 axons to the median eminence (Gill & Tsai, 2006; Ojeda, Lomniczi & Sandau, 2008). Therefore, understanding FGFR signaling can give further insight to the hypothalamic regulation of homeostasis.

Table 4 Qualitative comparison of Fgfr1 expression

Time point	Area	Fgfr1 expression in cell types	
		BLBP	GFAP	SOX2	DCX	NeuN	Olig2	
P7	DG	+ +	+ +	NA	+ +  +	+		
	CA		+ +	NA		+ +		
	SVZ	+ +  +	+	NA			+ +	
	Cortex	+ +  +	+ +	NA		+	+ +	
	Hypothalamus		+ +	NA				
1 month	DG		+ +	+ +	+ +			
	CA		+			+		
	SVZ		+ +	+ +				
	Cortex		+ +			+	+	
	Hypothalamus		+ +	+ +				
Notes.

NA not analyzed

Conclusion

FGFR1 has been implicated as having multiple functions in CNS development, homeostasis, and behavior, but defining the cellular basis of its functions depends on having a clear understanding of which cell types the Fgfr1 gene is expressed in, and when. The GENSAT project was envisioned as a resource that would provide the tools for such detailed studies (Heintz, 2004). Here, we have extended the previously published in situ based studies and online resources with a detailed examination of the tgFGR1-EGFPGP338Gsat line. Our data are congruent with in situ studies, but with the added feature of double immunofluorescence with glial and neuronal markers, and a quantification of the relative expression in glial versus neuronal cells in the young adult brain. We here find Fgfr1 promoter driven GFP expression in a variety of stem cells of the CNS including the young adult SVZ, young adult SGZ, cerebellar Bergmann Glia, and SOX2+ cells of the hypothalamus (summarized in Table 4). An additional study of Fgfr1 expression in embryonic and perinatal stem cells will be described elsewhere. We also find that Fgfr1 is expressed predominantly in glia in the young adult brain, although significant expression in DCX+ positive neuroblasts of the hippocampus was also observed. This was in stark contrast to DCX+ neuroblasts of the RMS. Our findings may shed light on the participation of FGF2/FGFR1 signaling in determining anxiety and mood, where a neuronal role for FGFR1 has been hypothesized (Turner, Watson & Akil, 2012a). Future studies are needed to determine whether Fgfr1 expression colocalizes with other markers, such as S100β, O4, and NG2 at embryonic and postnatal time points. Since Gfap is not expressed in all astrocytes, S100β (glial specific marker primarily expressed in astrocytes, but also in some ependymal cells) would be a good marker to further explore Fgfr1 expression. The tgFgfr1-EGFP+ model can also be used to study additional stages in development, or Fgfr1 expression after environmental manipulations previously shown to alter Fgfr1 expression, including animal models for induced depression such as social defeat stress.

Supplemental Information

Figure S1 Fgfr1 and GFAP expression in the DG and CA regions of the hippocampus of P7 mice

GFAP (red), and GFP immunostaining of the DG in P7 tgfgfr1-EGFP+ mice (A–C, and G–I, n = 3) and tgfgfr1-EGFP-controls (D–F, and J-L, n = 3). Comparison of the DG of the Hippocampus (A–C and D–F), and CA regions (G–I and J–L) revealed GFP+ staining in GFAP+ cells in the SGZ of the DG and in the CA region (D), as well as in white matter. DAPI, Blue channel. Dashed line indicates outer hippocampus/white matter boundary. Scale bars are 50 µm.

Click here for additional data file.

Figure S2 Fgfr1 and NeuN expression in CA region of the hippocampus P7 mice

NeuN (A, D) and GFP (B, E) immunostaining of the CA region in tgfgfr1-EGFP+ mice (A–C, n = 3) and tgfgfr1-EGFP- controls (D–F, n = 3). Neun+/GFP+ staining was observed in stratum pyramidale (SP) of the CA region. Dashed line indicated outer hippocampus/white matter boundary. Scale bars are 50 µm.

Click here for additional data file.

Figure S3 NeuN and GFAP staining in the cerebellum of tgfr1-EGFP+ mice at P7

NeuN+ granule cell layer neurons did not colocalize with GFP at P7 (A, low magnification, B, high magnification). GFAP+ glia of the cerebellum do colocalize with GFP at P7 (C, low magnification, D, high magnification). Arrow heads denote double labeled cells while small arrows denote GFAP+/GFP+ glial fibers. Scale bar is 50 µm in A, C, and 25 µm in B, D.

Click here for additional data file.

Figure S4 FGFR1 expression in the hippocampus at 1 Month in tgfgfr1-EGFP+ mice

Low magnification images of SOX2 (A) and SOX2/GFP colabeling (B) in the DG of tgfgfr1-EGFP+ mice. Low magnification images of DCX (C) and DCX/GFP colabeling (D) in the DG of tgfgfr1-EGFP+ mice. High Magnification images of GFAP (red)/NeuN (blue)/GFP triple staining in the DG (E) and CA (F). Scale bar is 50 µm in A–D, and 25 µm in (E-F).

Click here for additional data file.

Figure S5 Fgfr1 promoter driven GFP does not colocalize with Olig2+ cells in the hippocampus

OLIG2 staining in the hippocampus of control tgfgfr1-EGFP-mice (A–D) and tgfgfr1-EGFP+ mice (E–H). Scale bar is 50 µm.

Click here for additional data file.

Figure S6 FGFR1 expression in the hypothalamus

Fgfr1 promoter driven GFP in the hypothalamus of control tgfgfr1-EGFP-mice (A, B) and tgfgfr1-EGFP+ mice (C–F). OLIG2+ cells in the hypothalamus do not colocalize with Fgfr1 promoter driven GFP+ (C, D). BLBP+ cells near the third ventricle do colocolize with Fgfr1 promoter driven GFP (E), but do not colocalize with GFP in the medial eminence (F). Scale bar is 50 µm in A, C, E and 25 µm in B, D, F.

Click here for additional data file.

The authors wish to thank Darryl Williams, Deborah June Rogers, Lori Rubin and Marques Jackson for technical assistance, and Glen Watson and Caryl Chlan for manuscript review.

Additional Information and Declarations

Competing Interests

Author Contributions

Animal Ethics

Data Availability

The authors declare there are no competing interests.

Lisha Choubey conceived and designed the experiments, performed the experiments, analyzed the data, wrote the paper, prepared figures and/or tables, reviewed drafts of the paper.

Jantzen C. Collette performed the experiments, analyzed the data, wrote the paper, prepared figures and/or tables, reviewed drafts of the paper.

Karen Müller Smith conceived and designed the experiments, performed the experiments, analyzed the data, contributed reagents/materials/analysis tools, wrote the paper, prepared figures and/or tables, reviewed drafts of the paper.

The following information was supplied relating to ethical approvals (i.e., approving body and any reference numbers):

University of Louisiana at Lafayette IACUC committee APS number 2012-8717-046, 2013-8717-053, 2014-8717 040, 2015-8717-033.

The following information was supplied regarding data availability:

https://figshare.com/s/d16b14dd2e71525a7293.

https://figshare.com/s/36bc336d5b0ccb95efe8.

https://figshare.com/s/722ecee87a3f32427759.

https://figshare.com/s/f17d6889939bd680152d.

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
