# Peer review of "Quantitative assessment of fibroblast growth factor receptor 1 expression in neurons and glia"

_PeerJ, doi:10.7717/peerj.3173_

## Round 0.1 · original submission · Major Revisions

Dear Karen,

Although interesting and scientifically sound, the paper, in the present form, shows some elements of weakness. A more detailed analysis of the results, also supported by more representative and clear images, and a critical discussion are strongly recommended to effectively improve the quality of your manuscript.

·

Basic reporting

Major Points:

1 - In all of the Figs except Figure 8, the brightness of figures for all panels needs to be systematically increased, if possibly, equally for all. I am unsure of this reflects the reduced quality in the pdf, but looking at them from a sufficiently bright screen it is difficult to observe any details from 30 cm distance. Because of this, the conclusions made don’t appear obvious enough because one cannot tell colocalisation. This is the most important weakness of the study.

2 – In most of the Figs, it is difficult to observe the orientation or substructures of the forebrain. This would be highly aided if there was some more touch up to the Figs in terms of dashed lines indicated for where there is the ventricular space and perhaps another line, 70 ums in from the edge of the ependymal layer for where is the SVZ. Then labelling SVZ as done for SP for example. Some of the HP pictures are easier to follow than the SVZ pictures. This would help readers to digest the data better.

3 – In Fig 2 and 3 for example, pics containing the SVZ should be flipped over horizontally to be in the same direction and throughout for all figures. In these 2 Figs, the SVZ is at the right side and in Fig 3 on the left side. In f Please change these to make it consistent for readers.

4 – Arrows and arrowheads throughout are small. Should be enlarged.

5 – There are no confocal images to show convincingly colocalisation. I would recommend the authors to implement aspects of this if possible. But this is more of a personal preference. Although the Zeiss scope used is still useful.

6 – The amount of DAPI seen in the top and bottom panels of Fig 7 is very very low for an overview section, especially the lower panel. If adjusting the brightness does not amend this, I would recommend changing these pictures to show DAPI filling cells as expected all around the cortex. Amount of DAPI here seems too low and odd.

7 – The discussion is overly long for a descriptive study explaining the mouse lines GFP expression pattern. Much of the functional aspects should be summarized in the context of the useful of this mouse line for aiding which type of future studies it could benefit.

8 – In general, references and discussions to in situ experiments (becoming more obsolete) done elsewhere should be avoided with newer, sensitive and better RNA studies.

Minor point:
1 – Some minor typos remain in the manuscript. Pass the manuscript to colleagues once checking through. For example “Therefore, FGFR1 has important roles in glial function and development including stem cell proliferation, migration, morphology, and support of neural maturation..” – extra full stop.

2 – The use of Fortin et al 2005 as a citing reference for the opening line seems inappropriate and irrelevant since this study refers to a specific glial type. A better ref on more generic CNS study than a specific cell type is needed or even a more recent review is ideal.

3 - Some of the refs used in the next line are also are also well outdated. Instead, refs from Azim et al in Glia 2012 and work from Fernando de Castro lab is suited best and also in the discussion in the second paragraph. Some of the refs also in the next lines with multiple refs should be updated with newer refs as these ones. Please change these.

4 – Second last para of introduction. Some of the work from Bansal lab that has been cited is based on in vitro and at times newer in vivo work from the same lab discredits their earlier culture findings. OPCs or stem cells placed in vitro change by a few 1000 genes rapidly. This is also mentioned in the cited Cahoy study. References to in vitro should be removed if possible.

5 – Perhaps the introduction could be shortened as this study isn’t based on a functional study but for a mice mouse line. The last para shouldn’t be changed though.

6 – Fig 3 could benefit with having color indicators in like in the other Figs above the pictures.

7 – The statement “Postnatally, the SVZ of the lateral ventricles gives rise to olfactory bulb interneurons. GFP+ cells were observed in the SVZ, consistent with previous studies of postnatal Fgfr1 expression (Fagel et al., 2009).” - Is false. Postnatally, the SVZ, depending on which region is analyzed will give rise to all cortical astrocytes and most of the remaining oligodendroglia after birth come from the dorsal SVZ. The dorsal SVZ contributes to neurons about 45% of the time, with the rest being parenchymal glia or ependymal cells. This statement must be amended because postnatal stem cell activity is related to all neural cell types in the forebrain.

8 – The statement “However, DCX did not colocalize with GFP staining in the SVZ of the lateral ventricles (Fig. 3C), in contrast to what was observed in the DG of the hippocampus.” - Should be backed up better. The SGZ will generate mainly projection neurons whilst the SVZ makes generally much more interneurons. These 2 different neuronal classes generated in these 2 neurogenic regions could be dependent on the expression of many thousands of genes, including fgfrs.

9 – The statement and the lines below “We examined whether tgfgfr1-EGFP+ was expressed in oligodendrocytes and their precursors by staining for OLIG2 in the cortex.” – GFP in these overview pictures seems lower, again would have been benefited with high mag confocal pictures, but the lower level of GFP at P7 could reflect OPCs with the lag of GFP expression which can exist for days after initial fgfr1 expression in earlier cells. Since others have shown Fgfr1 is detected in Ascl1+ progenitors that make both OPS and neurons, GFP in Olig2 with the pictures provided seems that GFP is merely extended into progenies of progenitors which is common. Authors should adjust their statements here accordingly.

10 – This statement must be removed “NeuN+/GFP+ cells were not observed in the SVZ.” NeuN cells are never seen in the SVZ in any conditions.

11 – The statement “Therefore, the insertion of this transgene into the host DNA does not cause toxicity amongst the neuronal and glial cells observed.” – should not be in the discussion. Should be mentioned once only in the methods. Any other references to its lethality must be omitted. Gensat GFP mouse lines for standard reporter labelling are not known to affect mouse.

12 – It should be noted briefly in the discussion that postnatal pallial CD133+ neurogenic cells (where projection neurons and glial cells come from) have high levels of fgfr1 compared to the other cells in the SVZ with Azim et al 2012 Glia, which fits with why DCX+ cells have no fgfr1 expression. The expression levels have also been mapped to more pallial postnatal stem cells in a recent study by the same authors in Stem Cells. In addition, they should mention how fgfr1 is also enriched in subpopulations of stem cells of the adult SVZ throughout compared to other glial or parenchymal cell populations by citing Llorens-Bobadilla Cell Stem Cell 2015. The authors have to mention these and that perhaps focus on newer more sensitive gene expression profiling studies in replacing older references used in the manuscript. Would be ideal after this statement “Thus, astrocytes constitute a majority of Fgfr1 expressing cells in the adult brain. This idea is consistent with previous mRNA profiling of astrocytes in which Fgfr1 was identified as a gene with enriched expression in astrocytes ascompared to other cell types in the brain (Lovatt et al., 2007, Cahoy et al., 2008, Doyle et al. 2008).” and also to amend in conclusions “Here, we have extended the previously published in situ based studies and online resources with a detailed examination of the tgFGR1-EGFPGP338Gsat line.” with a text explaining…..for .e.g. “ with newer more sensitive transcriptional profiling expanding on these observations”.

Experimental design

No Comments

Validity of the findings

Some adjustments have to be made. See above.

·

Basic reporting

1. Overall the manuscript is well written, there are only few typos that should be corrected. See line 96; line 215; line 431; figure 5 legend line 11 (perfumed); table 1 "NueN". Line 424, it is not clear to me the meaning of "development of synaptic trasmission"
2. Abbreviation in the title should be avoided.
3. Introduction, although a bit long, is fine.
4. With regard to Figures presented in the manuscript, given that results of this study are all based on immunohistochemical analysis, representative images should be of higher quality, expecially those of SVZ and CA region (NeuN+ pyramidal neurons are hardly visible in Fig. 2A,D). Furthermore, low magnification images should be provided for all areas under study to allow their identification. Authors should also provide higher magnification images showing colocalization of GFP with the different cell markers.
All Figures should indicate colabeling with arrows or arrowheads.

Experimental design

1. Total number of animals should be provided in the Method section. Were all immunohistochemical analyses performed on the same animals? Sample size seems to be quite small (n=2,3).
2. Table 1 reports an antibody, Tbr2, that was not used in the study.

Validity of the findings

1. The Authors should provide more accurate description of their findings with regard to the brain area under study. In particular, the Authors refer generically to "CA region" of the hippocampus and to "brain cortex" without indicating any specific area. For the hippocampus, did the Authors pool data from CA1, CA2, CA3 regions? Did they obtain similar results in all CA regions? This should be assessed and clearly stated. The same point should be addressed for data regarding Fgfr1 expression in the cortex. An accurate analysis and description of the level of Fgfr1 expression in the different cortical areas should be provided. This information would strenghten the study providing further data to be discussed in the appropriate section.
2. In order to improve originality of the study and assess specificity of Fgfr1 expression, the Authors should also assess the level of Fgfr1 expression in "non canonical" regions, not described in previous study.
3. Discussion should be revised avoiding repetition of results and information already provided in the Introduction. Instead, the relevance of findings should be discussed with regard to the research question defined by the Authors: how does the understanding of which cell type express Fgfr1 elucidate its role in physiology and disease?
4. Authors should give less fragmentary and more clear interpretation of their results answering to questions like, for instance, is the expression of Fgfr1 a common feature of astrocytes? Is the level of expression in glial cells similar in different brain areas? Is Fgfr1 expressed ubiquitously? How would the neuron to glia signalling or the neurogenic process based on Fgf/Fgfr have a role in neuropsychiatric disorders? Is the expression of Fgfr1 developmentally regulated (comparison of data obtained at P7 and 1 month)?

Additional comments

The study by Choubey et al., is interesting and well written, however there are some weaknesses in the immunohistochemical analysis and discussion section that need to be properly addressed in order to enhance the quality of the manuscript for publication in PeerJ.

---

## Round 0.2 · Minor Revisions

For a better clarity of the manuscript, my advise is to amend the misspellings indicated by Reviewer 2 and to answer the points raised by Reviewer 1.

·

Basic reporting

I believe now the manuscript is almost ready for publication. I would prefer if the authors can enhance their images. I believe with the microscope they have used, they have been too modest in using the brightness and contrasts which I believe in many journals I have used allows.

Experimental design

No Comment

Validity of the findings

The findings are very sound and I could also acquire this mouse line in future.

Additional comments

Dear Lisha Choubey, Jantzen Collette and Karen Muller Smith,
It was a pleasure to review the manuscript and see the changes again now. I could see that the manuscript has been turned around for the better overall. The introduction and discussion has been improved.

I have used similar mouse lines such as Sox2-GFP, Nestin-GFP, GFAP-GFP, etc, and I feel this mouse line is very much undervalued. Indeed, it seems superior and has advantages to these mentioned mouse lines that a multitude of labs are using. Therefore this manuscript should be very well cited once this mouse line becomes distributed over time. I have some minor additions/points that the authors could/should/may implement.

1 – I would state in the abstract that fgfr1 is also involved in fate specification in the second line. Just to reflect the info in the introduction and since radial glia don’t proliferate so much compared to their early progenitors.

2 – Are the authors able to comment on if the GFP can be seen without using an antibody to enhance its signal? If no antibody is needed, this is a major selling point as it would allow various other experiments to be done without the need of an antibody and perhaps the authors could mention this.

Another half critical point is that:
1 – still with the images, I feel unless the brightness is at its highest on a PC screen, I am unable to make out the substructures of the SGZ/SVZ especially. In more periventricular regions, there is no need for this in this study. But in the germinal zones it seems its needed.

With all of the work done by the authors, it would be a shame if readers cant completely make out what is that. Even though I have worked on these germinal zones for 12 years now, without extra brightness again, I would feel that perhaps readers wont have the complete insight into the depth due to the resolution. I wonder if the editor will allow the readers to change the brightness/constrast for crisper figures systematically for all panels of all figures in the same way. Other journals do allow this. This is easily doable. Another way around this (and in addition), would be to perhaps insert in the main text what the error bars are describing rather than readers going into the figure legends. Having these small descriptions in both results section and figure legends will guide readers better.

2 – The labelling implemented in the figures should be better placed. For example, In figure 4, in the top panels, the DG should be placed directly where the DG is and same for the SVZ which is actually the corpus callosum. In the this panel too, the SVZ label could be placed inside the dotted lines/marked area. This is to aid readers to really what is what. In many panels of figures this should be in the right place. The same labels could also be placed in Figure 6 and 8 to avoid readers looking so close and by seeing the label in the right spot, it is obvious. It skips the need to read again the figure legends.

Otherwise, I am convinced that this mouse line will be a great tool for future studies in different labs in different fields. Therefore, a description of this mouse line such as this manuscript will be very important for other investigators to follow for various utilities and should be well read/cited.

I apologize for further requests, but it is merely to finalise a study that must be accepted. I hope I have been of assistance. Yours sincerely, Reviewer from the Neurogenesis field.

·

Basic reporting

No comment

Experimental design

No comment

Validity of the findings

No comment

Additional comments

The manuscript revision has addressed the main concerns made about the original manuscript. The Authors have improved the quality of previous figures and added new ones that allow a better identification of structures and labelings.
Comments given in the Results and Discussion have strengthened the Authors' conclusions and improved the global quality of the paper.
The manuscript is well written, however some misspellings are still present. See, for instance, line 180: "...so the cortex was counted in it’s entirety using stereology..." "It's" should be replaced by its. Please re-check carefully expecially the sentences added in the revised version.

---

## Round 0.3 · accepted · Accept

I wish to thank the Authors for their efforts to effectively improve the quality of the manuscript.